# StructPool: Structured Graph Pooling via Conditional Random Fields

**Hao Yuan**
Department of Computer Science & Engineering
Texas A&M University
College Station, TX 77843, USA
`hao.yuan@tamu.edu`

**Shuiwang Ji**
Department of Computer Science & Engineering
Texas A&M University
College Station, TX 77843, USA
`sji@tamu.edu`

## Abstract

Learning high-level representations for graphs is of great importance for graph analysis tasks. In addition to graph convolution, graph pooling is an important but less explored research area. In particular, most of existing graph pooling techniques do not consider the graph structural information explicitly. We argue that such information is important and develop a novel graph pooling technique, know as the StructPool, in this work. We consider the graph pooling as a node clustering problem, which requires the learning of a cluster assignment matrix. We propose to formulate it as a structured prediction problem and employ conditional random fields to capture the relationships among the assignments of different nodes. We also generalize our method to incorporate graph topological information in designing the Gibbs energy function. Experimental results on multiple datasets demonstrate the effectiveness of our proposed StructPool.

## 1 Introduction

Graph neural networks have achieved the state-of-the-art results for multiple graph tasks, such as node classification (Veličković et al., 2018; Gao & Ji, 2019b; Gao et al., 2018) and link prediction (Zhang & Chen, 2018; Cai & Ji, 2020). These results demonstrate the effectiveness of graph neural networks to learn node representations. However, graph classification tasks also require learning good graph-level representations. Since pooling operations are shown to be effective in many image and NLP tasks, it is natural to investigate pooling techniques for graph data (Yu & Koltun, 2016; Springenberg et al., 2014). Recent work extends the global sum/average pooling operations to graph models by simply summing or averaging all node features (Atwood & Towsley, 2016; Simonovsky & Komodakis, 2017). However, these trivial global pooling operations may lose important features and ignore structural information. Furthermore, global pooling are not hierarchical so that we cannot apply them where multiple pooling operations are required, such as Graph U-Net (Gao & Ji, 2019a). Several advanced graph pooling methods, such as SORTPOOL (Zhang et al., 2018), TOPKPOOL (Gao & Ji, 2019a), DIFFPOOL (Ying et al., 2018), and SAGPOOL (Lee et al., 2019) , are recently proposed and achieve promising performance on graph classification tasks. However, none of them explicitly models the relationships among different nodes and thus may ignore important structural information. We argue that such information is important and should be explicitly captured in graph pooling.

In this work, we propose a novel graph pooling technique, known as the StructPool, that formulates graph pooling as a structured prediction problem. Following DIFFPOOL (Ying et al., 2018), we consider graph pooling as a node clustering problem, and each cluster corresponds to a node in the new graph after pooling. Intuitively, two nodes with similar features should have a higher probability of being assigned to the same cluster. Hence, the assignment of a given node should depend on both the input node features and the assignments of other nodes. We formulate this as a structured prediction problem and employ conditional random fields (CRFs) (Lafferty et al., 2001) to capture such high-order structural relationships among the assignments of different nodes. In addition, we generalize our method by incorporating the graph topological information so that our method can control the clique set in our CRFs. We employ the mean field approximation to compute the assignments and describe how to incorporate it in graph networks. Then the networks can be

trained in an end-to-end fashion. Experiments show that our proposed STRUCTPOOL outperforms existing methods significantly and consistently. We also show that STRUCTPOOL incurs acceptable computational cost given its superior performance.

## 2 BACKGROUND AND RELATED WORK

### 2.1 GRAPH CONVOLUTIONAL NETWORKS

A graph can be represented by its adjacency matrix and node features. Formally, for a graph $G$ consisting of $n$ nodes, its topology information can be represented by an adjacency matrix $A \in \{0,1\}^{n \times n}$, and the node features can be represented as $X \in \mathbb{R}^{n \times c}$ assuming each node has a $c$-dimensional feature vector. Deep graph neural networks (GNNs) learn feature representations for different nodes using these matrices (Gilmer et al., 2017). Several approaches are proposed to investigate deep GNNs, and they generally follow a neighborhood information aggregation scheme (Gilmer et al., 2017; Xu et al., 2019; Hamilton et al., 2017; Kipf & Welling, 2017; Veličković et al., 2018). In each step, the representation of a node is updated by aggregating the representations of its neighbors. Graph Convolutional Networks (GCNs) are popular variants of GNNs and inspired by the first order graph Laplacian methods (Kipf & Welling, 2017). The graph convolution operation is formally defined as:

$$X_{i+1} = f(D^{-\frac{1}{2}} \hat{A} D^{-\frac{1}{2}} X_i P_i),$$ (1)

where $\hat{A} = A + I$ is used to add self-loops to the adjacency matrix, $D$ denotes the diagonal node degree matrix to normalize $\hat{A}$, $X_i \in \mathbb{R}^{n \times c_i}$ are the node features after $i^{th}$ graph convolution layer, $P_i \in \mathbb{R}^{c_i \times c_{i+1}}$ is a trainable matrix to perform feature transformation, and $f(\cdot)$ denotes a non-linear activation function. Then $X_i \in \mathbb{R}^{n \times c_i}$ is transformed to $X_{i+1} \in \mathbb{R}^{n \times c_{i+1}}$ where the number of nodes remains the same. A similar form of GCNs proposed in (Zhang et al., 2018) can be expressed as:

$$X_{i+1} = f(D^{-1} \hat{A} X_i P_i).$$ (2)

It differs from the GCNs in Equation (1) by performing different normalization and is a theoretically closer approximation to the Weisfeiler-Lehman algorithm (Weisfeiler & Lehman, 1968). Hence, in our models, we use the latter version of GCNs in Equation (2).

### 2.2 GRAPH POOLING

Several advanced pooling techniques are proposed recently for graph models, such as SORTPOOL, TOPKPOOL, DIFFPOOL, and SAGPOOL, and achieve great performance on multiple benchmark datasets. All of SORTPOOL (Zhang et al., 2018), TOPKPOOL (Gao & Ji, 2019a), and SAG-POOL (Lee et al., 2019) learn to select important nodes from the original graph and use these nodes to build a new graph. They share the similar idea to learn a sorting vector based on node representations using GCNs, which indicates the importance of different nodes. Then only the top $k$ important nodes are selected to form a new graph while the other nodes are ignored. However, the ignored nodes may contain important features and this information is lost during pooling. DIFFPOOL (Ying et al., 2018) treats the graph pooling as a node clustering problem. A cluster of nodes from the original graph are merged to form a new node in the new graph. DIFFPOOL proposes to perform GCNs on node features to obtain node clustering assignment matrix. Intuitively, the cluster assignment of a given node should depend on the cluster assignments of other nodes. However, DIFFPOOL does not explicitly consider such high-order structural relationships, which we believe are important for graph pooling. In this work, we propose a novel structured graph pooling technique, known as the STRUCTPOOL, for effectively learning high-level graph representations. Different from existing methods, our method explicitly captures high-order structural relationships between different nodes via conditional random fields. In addition, our method is generalized by incorporating graph topological information $A$ to control which node pairs are included in our CRFs.

### 2.3 INTEGRATING CRFS WITH GNNS

Recent work (Gao et al., 2019; Qu et al., 2019; Ma et al., 2019) investigates how to combine CRFs with GNNs. The CGNF (Ma et al., 2019) is a GNN architecture for graph node classification which explicitly models a joint probability of the entire set of node labels via CRFs and performs inference

via dynamic programming. In addition, the GMNN (Qu et al., 2019) focuses on semi-supervised object classification tasks and models the joint distribution of object labels conditioned on object attributes using CRFs. It proposes a pseudolikelihood variational EM framework for model learning and inference. Recent work (Gao et al., 2019) integrates CRFs with GNNs by proposing a CRF layer to encourage similar nodes to have similar hidden features so that similarity information can be preserved explicitly. All these methods are proposed for node classification tasks and the CRFs are incorporated in different ways. Different from existing work, our STRUCTPOOL is proposed for graph pooling operation and the energy is optimized via mean field approximation. All operations in our STRUCTPOOL can be realized by GNN operations so that our STRUCTPOOL can be easily used in any GNNs and trained in an end-to-end fashion.

## 3 STRUCTURED GRAPH POOLING

### 3.1 GRAPH POOLING VIA NODE CLUSTERING

Even though pooling techniques are shown to facilitate the training of deep models and improve their performance significantly in many image and NLP tasks (Yu & Koltun, 2016; Springenberg et al., 2014), local pooling operations cannot be directly applied to graph tasks. The reason is there is no spatial locality information among graph nodes. Global max/average pooling operations can be employed for graph tasks but they may lead to information loss, due to largely reducing the size of representations trivially. A graph $G$ with $n$ nodes can be represented by a feature matrix $X \in \mathbb{R}^{n \times c}$ and an adjacent matrix $A \in \{0, 1\}^{n \times n}$. Graph pooling operations aim at reducing the number of graph nodes and learning new representations. Suppose that graph pooling generates a new graph $\tilde{G}$ with $k$ nodes. The representation matrices of $\tilde{G}$ are denoted as $\tilde{X} \in \mathbb{R}^{k \times \tilde{c}}$ and $\tilde{A} \in \{0, 1\}^{k \times k}$. The goal of graph pooling is to learn relationships between $X$, $A$ and $\tilde{X}$, $\tilde{A}$. In this work, we consider graph pooling via node clustering. In particular, the nodes of the original graph $G$ are assigned to $k$ different clusters. Then each cluster is transformed to a new node in the new graph $\tilde{G}$. The clustering assignments can be represented as an assignment matrix $M \in \mathbb{R}^{n \times k}$. For hard assignments, $m_{i,j} \in \{0, 1\}$ denotes if node $i$ in graph $G$ belongs to cluster $j$. For soft assignments, $m_{i,j} \in [0, 1]$ denotes the probability that node $i$ in graph $G$ belongs to cluster $j$ and $\sum_j m_{i,j} = 1$. Then the new graph $\tilde{G}$ can be computed as

$$\tilde{X} = M^T X, \ \tilde{A} = g(M^T A M), \tag{3}$$

where $g(\cdot)$ is a function that $g(\tilde{a}_{i,j}) = 1$ if $\tilde{a}_{i,j} > 0$ and $g(\tilde{a}_{i,j}) = 0$ otherwise.

### 3.2 LEARNING CLUSTERING ASSIGNMENTS VIA CONDITIONAL RANDOM FIELDS

Intuitively, node features describe the properties of different nodes. Then nodes with similar features should have a higher chance to be assigned to the same cluster. That is, for any node in the original graph $G$, its cluster assignment should not only depend on node feature matrix $X$ but also condition on the cluster assignments of the other nodes. We believe such high-order structural information is useful for graph pooling and should be explicitly captured while learning clustering assignments. To this end, we propose a novel structured graph pooling technique, known as STRUCTPOOL, which generates the assignment matrix by considering the feature matrix $X$ and the relationships between the assignments of different nodes. We propose to formulate this as a conditional random field (CRF) problem. The CRFs model a set of random variables with a Markov Random Field (MRF), conditioned on a global observation (Lafferty et al., 2001). We formally define $Y = \{Y_1, \cdots, Y_n\}$ as a random field where $Y_i \in \{1, \cdots, k\}$ is a random variable. Each $Y_i$ indicates to which cluster the node $i$ is assigned. Here the feature representation $X$ is treated as global observation. We build a graphical model on $Y$, which is defined as $G'$. Then the pair $(Y, X)$ can be defined as a CRF, characterized by the Gibbs distribution as

$$P(Y|X) = \frac{1}{Z(X)} \exp\left(-\sum_{c \in C_{G'}} \psi_c(Y_c|X)\right), \tag{4}$$

where $c$ denotes a clique, $C_{G'}$ is a set of cliques in $G'$, $Z(X)$ is the partition function, and $\psi_c(\cdot)$ is a potential function induced by $c$ (Krähenbühl & Koltun, 2011; Lafferty et al., 2001). Then the Gibbs

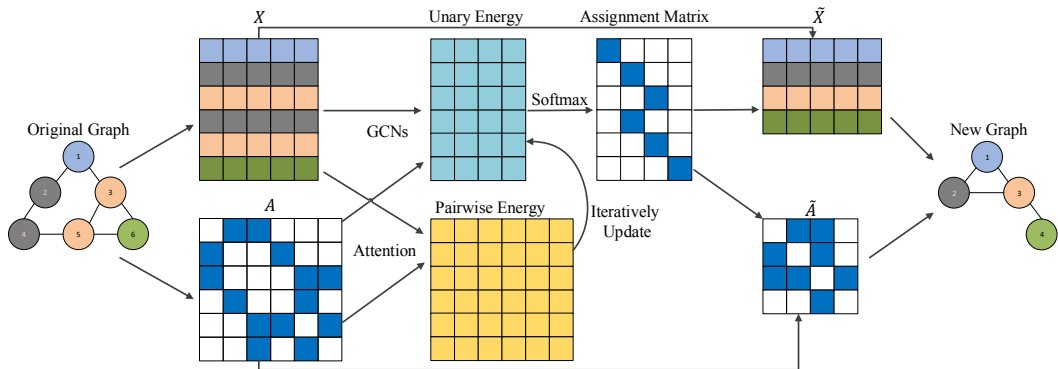

Figure 1: Illustrations of our proposed STRUCTPOOL. Given a graph with 6 nodes, the color of each node represents its features. We perform graph pooling to obtain a new graph with $k = 4$ nodes. The unary energy matrix can be obtained by multiple GCN layers using $X$ and $A$. The pairwise energy is measured by attention matrix using node feature $X$ and topology information $A$. Then by performing iterative updating, the mean field approximation yields the most probable assignment matrix. Finally, we obtain the new graph with 4 nodes, represented by $\tilde{X}$ and $\tilde{A}$.

energy function for an assignment $y = \{y_1, \cdots, y_n\}$ for all variables can be written as

$$E(y|X) = \sum_{c \in C_{G'}} \psi_c(y_c|X). \tag{5}$$

Finding the optimal assignment is equivalent to maximizing $P(Y|X)$, which can also be interpreted as minimizing the Gibbs energy.

### 3.3 GIBBS ENERGY WITH TOPOLOGY INFORMATION

Now we define the clique set $C_{G'}$ in $G'$. Similar to the existing CRF model (Krähenbühl & Koltun, 2011), we include all unary cliques in $C_{G'}$ since we need to measure the energy for assigning each node. For pairwise cliques, we generalize our method to control the pairwise clique set by incorporating the graph topological information $A$. We consider $\ell$-hop connectivity based on $A$ to define the pairwise cliques, which builds pairwise relationships between different nodes. Let $A^\ell \in \{0, 1\}^{n \times n}$ represent the $\ell$-hop connectivity of graph $G$ where $a_{i,j}^\ell = 1$ indicates node $i$ and node $j$ are reachable in $G$ within $\ell$ hops. Then we include all pairwise cliques $(i, j)$ in $C_{G'}$ if $a_{i,j}^\ell = 1$. Altogether, the Gibbs energy for a cluster assignment $y$ can be written as

$$E(y) = \sum_i \psi_u(y_i) + \sum_{i \neq j} \psi_p(y_i, y_j) a_{i,j}^\ell, \tag{6}$$

where $\psi_u(y_i)$ represents the unary energy for node $i$ to be assigned to cluster $y_i$. In addition, $\psi_p(y_i, y_j)$ is the pairwise energy, which indicates the energy of assigning node $i, j$ to cluster $y_i, y_j$ respectively. Note that we drop the condition information in Equation (6) for simplicity. If $\ell$ is large enough, our CRF is equivalent to the dense CRFs. If $\ell$ is equal to 1, we have $A^\ell = A$ so that only 1-hop information in the adjacent matrix is considered. These two types of energy can be obtained directly by neural networks (Zheng et al., 2015). Given the global observations $X$ and the topology information $A$, we employ multiple graph convolution layers to obtain the unary energy $\Psi_u \in \mathbb{R}^{n \times k}$. Existing work on image tasks (Krähenbühl & Koltun, 2011) proposes to employ Gaussian kernels to measure the pairwise energy. However, due to computational inefficiency, we cannot directly apply it to our CRF model. The pairwise energy proposed in (Krähenbühl & Koltun, 2011) can be written as

$$\psi_p(y_i, y_j) = \mu(y_i, y_j) \sum_{m=1}^{K} w^{(m)} k^{(m)}(x_i, x_j), \tag{7}$$

where $k^{(m)}(\cdot, \cdot)$ represents the $m^{th}$ Gaussian kernel, $x_i$ is the feature vector for node $i$ in $X$, $w^{(m)}$ denotes learnable weights, and $\mu(y_i, y_j)$ is a compatibility function that models the compatibility

---

**Algorithm 1** STRUCTPOOL

---

1: Given a graph $G$ with $n$ nodes represented by $X \in \mathbb{R}^{n \times c}$ and $A \in \{0,1\}^{n \times n}$, the goal is to obtain $\tilde{G}$ with $k$ nodes that $\tilde{X} \in \mathbb{R}^{k \times \tilde{c}}$ and $\tilde{A} \in \{0,1\}^{k \times k}$. The $\ell$-hop connectivity matrix $A^\ell$ can be easily obtained from $A$.
2: Perform GCNs to obtain unary energy matrix $\Psi_u \in \mathbb{R}^{n \times k}$.
3: Initialize that $Q(i,j) = \frac{1}{Z_i} \exp\left(\Psi_u(i,j)\right)$ for all $0 \le i \le n$ and $0 \le j \le k$.
4: **while** not converged **do**
5:      Calculate attention map $W$ that $w_{i,j} = \frac{x_i^T x_j}{\sum_{m \neq i} x_i^T x_m} a_{i,j}^\ell$ for all $i \neq j$ and $0 \le i,j \le n$.
6:      Message passing that $\tilde{Q}(i,j) = \sum_{m \neq i} w_{i,m} Q(m,j)$.
7:      Compatibility transform that $\hat{Q}(i,j) = \sum_m \mu(m,j) \tilde{Q}(i,m)$.
8:      Local update that $\bar{Q}(i,j) = \Psi_u(i,j) - \hat{Q}(i,j)$.
9:      Perform normalization that $Q(i,j) = \frac{1}{Z_i} \exp\left(\bar{Q}(i,j)\right)$ for all $i$ and $j$.
10: **end while**
11: For soft assignments, the assignment matrix is $M = \text{softmax}(Q)$.
12: For hard assignments, the assignment matrix is $M = \text{argmax}(Q)$ for each row.
13: Obtain new graph $\tilde{Q}$ that $\tilde{X} = M^T X, \tilde{A} = g(M^T A M)$.

---

between different assignment pairs. However, it is computationally inefficient to accurately compute the outputs of Gaussian kernels, especially for graph data when the feature vectors are high-dimensional. Hence, in this work, we propose to employ the attention matrix as the measurement of pairwise energy. Intuitively, Gaussian kernels indicate how strongly different feature vectors are connected with each other. Similarly, the attention matrix reflects similarities between different feature vectors but with a significantly less computational cost. Specifically, each feature vector $x_i$ is attended to any other feature vector $x_j$ if the pair $(i,j)$ is existing in clique set $C_{G'}$. Hence, the pairwise energy can be obtained by

$$\psi_p(y_i, y_j) = \mu(y_i, y_j) \frac{x_i^T x_j}{\sum_{k \neq i} x_i^T x_k}, \tag{8}$$

It can be efficiently computed by matrix multiplication and normalization. Minimizing the Gibbs energy in Equation (6) results in the most probable cluster assignments for a given graph $G$. However, such minimization is intractable, and hence a mean field approximation is proposed (Krähenbühl & Koltun, 2011), which is an iterative updating algorithm. We follow the mean-field approximation to obtain the most probable cluster assignments. Altogether, the steps of our proposed STRUCT-POOL are shown in Algorithm 1. All operations in our proposed STRUCTPOOL can be implemented as GNN operations, and hence the STRUCTPOOL can be employed in any deep graph model and trained in an end-to-end fashion. The unary energy matrix can be obtained by stacking several GCN layers, and the normalization operations (step 3&9 in Algorithm 1) are equivalent to softmax operations. All other steps can be computed by matrix computations. It is noteworthy that the compatibility function $\mu(y_i, y_j)$ can be implemented as a trainable matrix $\mathcal{N} \in \mathbb{R}^{k \times k}$, and automatically learned during training. Hence, no prior domain knowledge is required for designing the compatibility function. We illustrate our proposed STRUCTPOOL in Figure 1 where we perform STRUCTPOOL on a graph $G$ with 6 nodes, and obtain a new graph $\tilde{G}$ with 4 nodes.

### 3.4 Computational Complexity Analysis

We theoretically analyze the computational efficiency of our proposed STRUCTPOOL. Since computational efficiency is especially important for large-scale graph datasets, we assume that $n > k, c, \tilde{c}$. The computational complexity of one GCN layer is $\mathcal{O}(n^3 + n^2 c + n c \tilde{c}) \approx \mathcal{O}(n^3)$. Assuming we employ $i$ layers of GCNs to obtain the unary energy, its computational cost is $\mathcal{O}(i n^3)$. Assuming there are $m$ iterations in our updating algorithm, the computational complexity is $\mathcal{O}(m(n^2 c + n^2 k + n k^2)) \approx \mathcal{O}(m n^3)$. The final step for computing $\tilde{A}$ and $\tilde{X}$ takes $\mathcal{O}(n k c + n^2 k + n k^2) \approx \mathcal{O}(n^3)$ computational complexity. Altogether, the complexity STRUCT-POOL is $\mathcal{O}((m+i)n^3)$, which is close to the complexity of stacking $m + i$ layers of GCNs.

Table 1: Classification results for six benchmark datasets. Note that none of these deep methods can outperform the traditional method WL on COLLAB. We believe the reason is the graphs in COLLAB only have single-layer structures while deep models are too complex to capture them.

| Method | Dataset | | | | | |
|---|---|---|---|---|---|---|
| | ENZYMES | D&D | COLLAB | PROTEINS | IMDB-B | IMDB-M |
| GRAPHLET | 41.03 | 74.85 | 64.66 | 72.91 | - | - |
| SHORTEST-PATH | 42.32 | 78.86 | 59.10 | 76.43 | - | - |
| WL | 53.43 | 78.34 | **78.61** | 74.68 | - | - |
| PATCHYSAN | - | 76.27 | 72.60 | 75.00 | 71.00 | 45.23 |
| DCNN | - | 58.09 | 52.11 | 61.29 | 49.06 | 33.49 |
| DGK | - | - | 73.09 | 71.68 | 66.96 | 44.55 |
| ECC | 53.50 | 72.54 | 67.79 | 72.65 | - | - |
| GRAPHSAGE | 54.25 | 75.42 | 68.25 | 70.48 | - | - |
| SET2SET | 60.15 | 78.12 | 71.75 | 74.29 | - | - |
| DGCNN | 57.12 | 79.37 | 73.76 | 75.54 | 70.03 | 47.83 |
| DIFFPOOL | 62.53 | 80.64 | 75.48 | 76.25 | - | - |
| STRUCTPOOL | **63.83** | **84.19** | 74.22 | **80.36** | **74.70** | **52.47** |

### 3.5 DEEP GRAPH NETWORKS FOR GRAPH CLASSIFICATION

In this section, we investigate graph classification tasks which require both good node-level and graph-level representations. For most state-of-the-art deep graph classification models, they share a similar pipeline that first produces node representations using GNNs, then performs pooling operations to obtain high-level representations, and finally employs fully-connected layers to perform classification. Note that the high-level representations can be either a vector or a group of $k$ vectors. For a set of graphs with different node numbers, with a pre-defined $k$, our proposed STRUCTPOOL can produce $k$ vectors for each graphs. Hence, our method can be easily generalized and coupled to any deep graph classification model. Specially, our model for graph classification is developed based on DGCNN (Zhang et al., 2018). Given any input graph, our model first employs several layers of GCNs (Equation (2)) to aggregate features from neighbors and learn representations for nodes. Next, we perform one STRUCTPOOL layer to obtain $k$ vectors for each graph. Finally, 1D convolutional layers and fully-connected layers are used to classify the graph.

## 4 EXPERIMENTAL STUDIES

### 4.1 DATASETS AND EXPERIMENTAL SETTINGS

We evaluate our proposed STRUCTPOOL on eight benchmark datasets, including five bioinformatics protein datasets: ENZYMES, PTC, MUTAG, PROTEINS (Borgwardt et al., 2005), D&D (Dobson & Doig, 2003), and three social network datasets: COLLAB (Yanardag & Vishwanathan, 2015b), IMDB-B, IMDB-M (Yanardag & Vishwanathan, 2015a). Most of them are relatively large-scale and hence suitable for evaluating deep graph models. We report the statistics and properties of them in Supplementary Table 6. Please see the Supplementary Section A for experimental settings.

We compare our method with several state-of-the-art deep GNN methods. PATCHYSAN (Niepert et al., 2016) learns node representations and a canonical node ordering to perform classification. DCNN (Atwood & Towsley, 2016) learns multi-scale substructure features by diffusion graph convolutions and performs global sum pooling. DGK (Yanardag & Vishwanathan, 2015a) models latent representations for sub-structures in graphs, which is similar to learn word embeddings. ECC (Simonovsky & Komodakis, 2017) performs GCNs conditioning on both node features and edge information and uses global sum pooling before the final classifier. GRAPHSAGE (Hamilton et al., 2017) is an inductive framework which generates node embeddings by sampling and aggregating features from local neighbors, and it employs global mean pooling. SET2SET (Vinyals et al., 2015) proposes an aggregation method to replace the global pooling operations in deep graph networks. DGCNN (Zhang et al., 2018) proposes a pooling strategy named SORTPOOL which sorts all nodes

Table 2: Comparisons between different pooling techniques under the same framework.

| Method | Dataset | | | | | |
|---|---|---|---|---|---|---|
| | ENZYMES | D&D | COLLAB | PROTEINS | IMDB-B | IMDB-M |
| SUM POOL | 47.33 | 78.72 | 69.45 | 76.26 | 51.69 | 42.76 |
| SORTPOOL | 52.83 | 80.60 | 73.92 | 76.83 | 70.00 | 46.26 |
| TOPK POOL | 53.67 | 81.71 | 73.34 | 77.47 | 72.80 | 49.00 |
| DIFFPOOL | 60.33 | 80.94 | 71.78 | 77.74 | 72.40 | 50.13 |
| SAGPOOL | **64.17** | 81.03 | 73.28 | 78.82 | 73.40 | 51.13 |
| STRUCTPOOL | 63.83 | **84.19** | **74.22** | **80.36** | **74.70** | **52.47** |

by learning and selects the first $k$ nodes to form a new graph. DIFFPOOL (Ying et al., 2018) is built based on GRAPHSAGE architecture but with their proposed differentiable pooling. Note that for most of these methods, pooling operations are employed to obtain graph-level representations before the final classifier. In addition, we compare our STRUCTPOOL with three graph kernels: Graphlet (Shervashidze et al., 2009), Shortest-path (Borgwardt & Kriegel, 2005), and Weisfeiler-Lehman subtree kernel (WL) (Weisfeiler & Lehman, 1968).

## 4.2 CLASSIFICATION RESULTS

We evaluate our proposed method on six benchmark datasets and compare with several state-of-the-art approaches. The results are reported in Table 1 where the best results are shown in bold and the second best results are shown with underlines. For our STRUCTPOOL, we perform 10-fold cross validations and report the average accuracy for each dataset. The 10-fold splitting is the same as DGCNN. For all comparing methods, the results are taken from existing work (Ying et al., 2018; Zhang et al., 2018). We can observe that our STRUCTPOOL obtains the best performance on 5 out of 6 benchmark datasets. For these 5 datasets, the classification results of our method are significantly better than all comparing methods, including advanced models DGCNN and DIFFPOOL. Notably, our model outperforms the second-best performance by an average of 3.58% on these 5 datasets. In addition, the graph kernel method WL obtains the best performance on COLLAB dataset and none of these deep models can achieve similar performance. Our model can obtain competitive performance compared with the second best model. This is because many graphs in COLLAB only have simple structures and deep models may be too complex to capture them.

## 4.3 COMPARISONS OF DIFFERENT POOLING METHODS

To demonstrate the effectiveness of our proposed pooling technique, we compare different pooling techniques under the same network framework. Specifically, we compare our STRUCTPOOL with the global sum pool, SORTPOOL, TOPKPOOL, DIFFPOOL, and SAGPOOL. All pooling methods are employed in the network framework introduced in Section 3.5. In addition, the same 10-fold cross validations from DGCNN are used for all pooling methods. We report the results in Table 2 and the best results are shown in bold. Obviously, our method achieves the best performance on five of six datasets, and significantly outperforms all comparing pooling techniques. For the dataset EN-ZYMES, our obtained result is competitive since SAGPOOL only slightly outperforms our proposed method by 0.34%. Such observations demonstrate the structural information in graphs is useful for graph pooling and the relationships between different nodes should be explicitly modeled.

## 4.4 STUDY OF COMPUTATIONAL COMPLEXITY

As mentioned in Section 3.4, our proposed STRUCTPOOL yields $\mathcal{O}((m + i)n^3)$ computational complexity. The complexity of DIFFPOOL is $\mathcal{O}(jn^3)$ if we assume it employs $j$ layers of GCNs to obtain the assignment matrix. In our experiments, $i$ is usually set to 2 or 3 which

Table 3: The prediction accuracy with different iteration number $m$.

| Dataset | $m = 1$ | $m = 3$ | $m = 5$ | $m = 10$ |
|---|---|---|---|---|
| ENZYMES | 62.67 | 63.00 | **63.83** | 63.50 |
| D&D | 82.82 | 83.08 | 83.59 | **84.19** |
| PROTEINS | 80.09 | 80.00 | **80.18** | **80.18** |

is much smaller than $n$. We conduct experiments to show how different iteration number $m$ affects the prediction accuracy and the results are reported in Table 3. Note that we employ the dense CRF form for all different $m$. We can observe that the performance generally increases with $m$ increasing, especially for large-scale dataset D&D. We also observe $m = 5$ is a good trade-off between time complexity and prediction performance. Notably, our method can even outperform other approaches when $m = 1$. Furthermore, we evaluate the running time of our STRUCTPOOL and compare it with DIFFPOOL. For 500 graphs from large-scale dataset D&D, we set $i = j = 3$ and show the averaging time cost to perform pooling for each graph. The time cost for DIFFPOOL is 0.042 second, while our STRUCTPOOL takes 0.049 second, 0.053 second and 0.058 second for $m = 1$, $m = 3$, $m = 5$ respectively. Even though our STRUCTPOOL has a relatively higher computational cost, it is still reasonable and acceptable given its superior performance.

## 4.5 EFFECTS OF TOPOLOGY INFORMATION

Next, we conduct experiments to show how the topology information $A^\ell$ affects the prediction performance. We evaluate our STRUCTPOOL with different $\ell$ values and report the results in Table 4. Note that when $\ell$ is large enough, our STRUCTPOOL considers all pairwise relationships between all nodes, and it is equiva-

Table 4: The prediction accuracy using different $A^\ell$ in STRUCTPOOL.

| Dataset | $\ell = 1$ | $\ell = 5$ | $\ell = 10$ | $\ell = 15$ | DENSE |
|---|---|---|---|---|---|
| IMDB-B | 74.60 | 74.40 | 74.30 | **74.70** | **74.70** |
| IMDB-M | 51.53 | 51.67 | 52.00 | 51.96 | **52.47** |
| PROTEINS | 79.73 | 79.61 | 79.83 | **80.36** | 80.18 |

lent to the dense CRF. For the datasets IMDB-M and PROTEINS, we can observe that the prediction accuracies are generally increasing with the increasing of $\ell$. With the increasing of $\ell$, more pairwise relationships are considered by the model, and hence it is reasonable to obtain better performance. In addition, for the dataset IMDB-B, the results remain similar with different $\ell$, and even $\ell = 1$ yields competitive performance with dense CRF. It is possible that 1-hop pairwise relationships are enough to learn good embeddings for such graph types. Overall, dense CRF consistently produces promising results and is a proper choice in practice.

## 4.6 GRAPH ISOMORPHISM NETWORKS WITH STRUCTPOOL

Recently, Graph Isomorphism Networks (GINs) are proposed and shown to be more powerful than traditional GNNs (Xu et al., 2019). To demonstrate the

Table 5: Comparisons with Graph Isomorphism Networks.

| Dataset | PTC | IMDB-B | MUTAG | COLLAB | IMDB-M |
|---|---|---|---|---|---|
| GINs | 64.60 | 75.10 | 89.40 | 80.20 | 52.30 |
| OURS | **73.46** | **78.50** | **93.59** | **84.06** | **54.60** |

effectiveness of our STRUCTPOOL and show its generalizability, we build models based on GINs and evaluate their performance. Specifically, we employ GINs to learn node representations and perform one layer of the dense form of our STRUCTPOOL, followed by 1D convolutional layers and fully-connected layers as the classifier. The results are reported in the Table 5, where we employ the same 10-fold splitting as GINs (Xu et al., 2019) and the GIN results are taken from its released results. These five datasets include both bioinformatic data and social media data, and both small-scale data and large-scale data. Obviously, incorporating our proposed STRUCTPOOL in GINs consistently and significantly improves the prediction performance. It leads to an average of 4.52% prediction accuracy improvement, which is promising.

## 5 CONCLUSIONS

Graph pooling is an appealing way to learn good graph-level representations, and several advaned pooling techiques are proposed. However, none of existing graph pooling techniques explicitly considers the relationship between different nodes. We propose a novel graph pooling technique, known as STRUCTPOOL, which is developed based on the conditional random fields. We consider the graph pooling as a node clustering problem and employ the CRF to build relationships between the assignments of different nodes. In addition, we generalize our method by incorporating the graph topological information so that our method can control the pairwise clique set in our CRFs. Finally,

we evaluate our proposed STRUCTPOOL on several benchmark datasets and our method can achieve new state-of-the-art results on five out of six datasets.

## ACKNOWLEDGEMENT

This work was supported in part by National Science Foundation grants DBI-1661289 and IIS-1908198.

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

# A  APPENDIX

## A.1  DATASETS AND EXPERIMENTAL SETTINGS

Table 6: Statistics and properties of eight benchmark datasets.

|  | Dataset | | | |
|---|---|---|---|---|
|  | ENZYMES | D&D | COLLAB | PROTEINS |
| # of Edges (avg) | 124.20 | 1431.3 | 2457.78 | 72.82 |
| # of Nodes (avg) | 32.63 | 284.32 | 74.49 | 39.06 |
| # of Graphs | 600 | 1178 | 5000 | 1113 |
| # of Classes | 6 | 2 | 3 | 2 |

|  | Dataset | | | |
|---|---|---|---|---|
|  | IMDB-B | IMDB-M | PTC | MUTAG |
| # of Edges (avg) | 96.53 | 65.94 | 14.69 | 19.79 |
| # of Nodes (avg) | 19.77 | 13.00 | 14.30 | 17.93 |
| # of Graphs | 1000 | 1500 | 344 | 188 |
| # of Classes | 2 | 3 | 2 | 2 |

We report the statistics and properties of eight benchmark datasets in Supplementary Table 6. For our STRUCTPOOL, we implement our models using Pytorch (Paszke et al., 2017) and conduct experiments on one GeForce GTX 1080 Ti GPU. The model is trained using Stochastic gradient descent (SGD) with the ADAM optimizer (Kingma & Ba, 2014). For the models built on DGCNN (Zhang et al., 2018) in Section 4.2, 4.3, 4.4, 4.5, we employ GCNs to obtain the node features and the unary energy matrix. All experiments in these sections perform 10-fold cross validations and we report the averaging results. The 10-fold splitting is exactly the same as DGCNN (Zhang et al., 2018). For the non-linear function, we employ tanh for GCNs and relu for 1D convolution layers. For the models built on GINs in Section 4.6, we employ GINs to learn node features and unary energy. Here the 10-fold splitting is exactly the same as GINs. We employ relu for all layers as the non-linear function. For all models, 1D convolutional layers and fully-connected layers are used after our STRUCTPOOL. Hard clustering assignments are employed in all experiments.

## A.2  EFFECTS OF PAIRWISE ENERGY

Table 7: Comparison with the baseline which excludes pairwise energy.

| Dataset | ENZYMES | D&D | COLLAB | PROTEINS | IMDB-B | IMDB-M |
|---|---|---|---|---|---|---|
| BASELINE | 60.83 | 81.30 | 70.58 | 78.18 | 72.40 | 50.13 |
| OURS | **63.83** | **84.19** | **74.22** | **80.36** | **74.70** | **52.47** |

We conduct experiments to show the importance of the pairwise energy. If the pairwise energy is removed, the relations between different node assignments are not explicitly considered. Then the method is similar to the DIFFPOOL. We compare our method with such a baseline that removes the pairwise energy. Experimental results are reported in Table 7. The network framework is the same as introduced in Section 3.5 and the same 10-fold cross validations from DGCNN are used. Obviously, our proposed method consistently and significantly outperforms the baseline which excludes pairwise energy. It indicates the importance and effectiveness of incorporating pairwise energy and considering high-order relationships between different node assignments.

## A.3  STUDY OF HIERARCHICAL NETWORK STRUCTURE

To demonstrate how the network depth and multiple pooling layers affects the prediction performance, we conduct experiments to evaluate different hierarchical network structures. We first define a network block contains two GCN layers and one STRUCTPOOL layer. Then we compare three

Table 8: Comparison with different hierarchical network structures.

| Dataset | 1 BLOCK | 2 BLOCKS | 3 BLOCKS |
|---------|---------|----------|----------|
| PROTEINS | 79.73 | 77.42 | 74.95 |
| D&D | 81.87 | 83.59 | 81.63 |

different network settings: 1 block with the final classifier, 2 blocks with the final classifier, and 3 blocks with the final classifier. The results are reported in Table 8. For the dataset Proteins, we observe that the network with one block can obtain better performance than deeper networks. We believe the main reason is dataset Proteins is a small-scale dataset with an average number of nodes equal to 39.06. A relatively simpler network is powerful enough to learn its data distribution while stacking multiple GCN layers and pooling layers may lead to a serious overfitting problems. For the dataset D&D, the network with 2 blocks performs better than the one with 1 block. Since D&D is relatively large scale, stacking 2 blocks increases the power of network and hence increases the performance. However, going very deep, e.g., stacking 3 blocks, will cause the overfitting problem.

## A.4 STUDY OF GRAPH POOLING RATE

Table 9: Comparison with different pooling rates.

| | $r = 0.1$ | $r = 0.3$ | $r = 0.5$ | $r = 0.7$ | $r = 0.9$ |
|-----|-----------|-----------|-----------|-----------|-----------|
| $k$ | 91 | 160 | 241 | 331 | 503 |
| ACC | 80.77 | 81.53 | 81.53 | 81.97 | 80.68 |

We follow the DGCNN (Zhang et al., 2018) to select the number of clusters $k$. Specifically, we use a pooling rate $r \in (0, 1)$ to control $k$. Then $k$ is set to an integer so that $r \times 100\%$ of graphs have nodes less than this integer in the current dataset. As suggested in DGCNN, generally, $r = 0.9$ is a proper choice for bioinformatics datasets and $r = 0.6$ is good for social network datasets. In addition, we conduct experiments to show the performance with the respect to different $r$ values. We set $r = 0.1, 0.3, 0.5, 0.7, 0.9$ to evaluate the performance on a large-scale social network dataset D&D. The average number of nodes in dataset D&D is 284.32 and the maximum number of nodes is 5748. The results are reported in Table 9 where the first row shows different pooling rates, the second row reports the corresponding $k$ values and the final row shows the results. For simplicity, we employ the network structure with 1 block and a final classifier (as defined in Section A.3). We can observe that the performance drops when $r, k$ is relatively large or small. In addition, the model can obtain competitive performance when $r$ is set to a proper range, for example, $r \in [0.3, 0.7]$ for dataset D&D.

