# OpenReview forum: "StructPool: Structured Graph Pooling via Conditional Random Fields"
_ICLR.cc/2020/Conference — Accept (Poster)_

### Official Review · AnonReviewer1 · 2019-10-22
**Official Blind Review #1**

**Rating:** 6

**Review:**

Strength:
-- An interesting idea to use CRF idea to cluster the nodes on a graph for pooling purpose
-- The paper is well written and easy to follow
--  On a few datasets and task, the proposed method works pretty well

Weakness:
-- The computational complexity of the proposed algorithm is on(n^3), which is too expensive

This paper studied to use CRF to define the cluster assignment of the nodes  for graph pooling, which model the dependency of the cluster assignments between the nodes. Experiments on a few data sets and experiments prove the effectiveness of the proposed approach over competitive baselines.

Although the proposed approach is interesting and works well on different datasets,  one strong concern I have is the high computational complexity of the algorithm, which is as high as o(n^3) (n is the number of nodes). This limits the applicability of the algorithm to large graphs.  Besides, I also have the following questions:
(1) In the experiments, I assume one CRF pooling layer is used. What would be the results if multiple layers are used?

(2) How did you select the number of clusters K? How would the performance change w.r.t. different K?


**Experience Assessment:**

I have published in this field for several years.

**Review Assessment: Checking Correctness Of Derivations And Theory:**

I carefully checked the derivations and theory.

**Review Assessment: Checking Correctness Of Experiments:**

I carefully checked the experiments.

**Review Assessment: Thoroughness In Paper Reading:**

I read the paper thoroughly.

---

> ### Author Response · Authors · 2019-11-11
> **Our response to Official Blind Review #1**
>
> Thanks for your constructive comments. We have revised our paper and please kindly check it.
>
> 1. First of all, we wish to point out the fact that our proposed method shares the same o(n^3) complexity with many advanced graph pooling techniques, such as SortPool [1], DiffPool [2] and SAGPool [3]. The main reason is that all of these methods involve GCN computations in the pooling and the computational complexity for 1-layer GCN is o(n^3). Thus, any GCN based pooling method will have the o(n^3) computational complexity.
> In terms of computational complexity, adding our Structpool layer is similar to stacking GCN layers. We analyze the computational complexity in Section 3.4 and 4.4. For 500 graphs from large-scale dataset D&D, the averaging time cost to perform pooling for each graph for Diffpool is 0.042s while our method takes 0.049 seconds, 0.053 second and 0.058 second for different numbers of iterations in mean-field approximation (m= 1,3,5). Our computation time is slightly higher but given our superior performance, we believe our cost is reasonable and acceptable. Note that we also show that even with m=1, our method yields very good performance.
>
> 2. Thanks for the suggestion. We conduct experiments to compare single pooling layer vs multiple pooling layers. Let’s define that 1 block contains 2 GCNs and 1 pooling. Then we compare three different models: 1 block + classifier, 2 blocks + classifier, and 3 blocks + classifier. We evaluate these three models on a large-scale dataset D&D and a small dataset Proteins. The results are reported in Appendix A.3. For the dataset Proteins, we observe that the network with one block can obtain better performance than deeper networks. We believe the main reason is dataset Proteins is a small-scale dataset with an average number of nodes equal to 39.06. A relatively simpler network is powerful enough to learn its data distribution while stacking multiple GCN layers and pooling layers leads to a serious overfitting problem. For the dataset D\&D, the network with 2 blocks performs better than the one with 1 block. Since D\&D is relatively large scale, stacking 2 blocks increases the power of the network and hence increases the performance. However, going very deep, e.g., stacking 3 blocks, will cause the overfitting problem.
>
> 3. We follow the DGCNN paper [1] to select the number of clusters $k$. Specifically, we use a pooling rate $(0<r<1)$ to control $k$. Then $k$ is set to an integer so that $r*100\%$ of graphs have nodes less than this integer in the current dataset. As suggested in DGCNN, generally, $r=0.9$ is a proper choice for bioinformatics datasets and $r=0.6$ is good for social network datasets. In addition, we conduct experiments to show the performance w.r.t. different $k$ values. We set $r= 0.1, 0.3, 0.5, 0.7, 0.9$ to evaluate the performance on a large-scale social network dataset D&D. The results are reported in Appendix A.4. We observe that the performance drops when $r, k$ is too large or small. In addition, we expect the model to have competitive performance when $r$ is set to a proper range, for example, $r\in[0.3,0.7]$ for dataset D&D.
>
> [1]. Zhang et al., An End-to-End Deep Learning Architecture for Graph Classification, AAAI 2018
> [2]. Ying et al., Hierarchical Graph Representation Learning with Differentiable Pooling, NIPS 2018
> [3]. Lee et al., Self-Attention Graph Pooling. ICML 2019.

---

> > ### Comment · AnonReviewer1 · 2019-11-15
> > **Response**
> >
> > The rebuttal clarifies my concerns and I would like to raise my ratings.

---

### Official Review · AnonReviewer2 · 2019-10-25
**Official Blind Review #2**

**Rating:** 6

**Review:**

The authors here introduces a novel  graph pooling technique called StructPool that uses the underlying graph’s structural information to behave as a node clustering algorithm and learns a node clustering matrix.

Graph level classification requires learning good graph level representation, especially for  aggregating low level information for high . Recent work in pooling does not take advantage of important structural information of the relationship between different  nodes. Here, the authors formulate  graph pooling as a structured prediction problem, control clique set in the CRFs and use mean filed approximation to calculate assignments.

A cluster assignment matrix assigns each node in the original graph to a cluster in the new graph. The assignment not only depends on the node features but also on the cluster assignment of the other nodes. The authors therefore draw connection with finding the optimal assignment to minimizing the Gibbs energy. The authors propose to learn clustering assignment via CRF conditioned on the global feature representation of the nodes.

The unary potentials of the cliques are computed used the GCN to measure energy of each node. The novelty in accommodating topology information is in using l hop connectivity based on adjacency A to define pairwise cliques thus building pairwise relationship between pairs of nodes thus allowing the Gibbs energy formulation of the cluster assignment thereby using GCN to also compute this pairwise energy.

I have  a few questions as below:
I think the authors can better elucidate the motivation for using  the attention matrix over Gaussian kernels to measure pairwise energy in section 3.3; an  empirical experiment for drawing comparison wrt to the computational time and number of feature dimensions on a toy problem seems important.
How is the computation  of the unary potential and pairwise energy influenced by the connectivity of the graph G for the datasets considered? It would be interesting to see how the pairwise energy, unary energy varies over different layers of GCNs.
Further, how is the cluster assignment affected by the l-hop connectivity?
Is there a notion of the minimum value of ‘k’ in the context of convergence?
What happens in case of very different graph features, or structural assumptions where the cliques are not enforced?
Is there a notion of how the method performs on datasets with a high percentage of isomorphism bias: repeating instances or repeating instances with different labels?
It will be interesting to see a discussion on how the performance varies with respect to the depth of the overall architecture,  positioning of the structpool and some results on how effective they are on  hierarchical features and multiple pooling ops as in architectures such as Graph UNet.
Avoid repetition in 2.2 Related work section and in other sections throughout. Otherwise, the paper is rather well written and has clarity.

**Experience Assessment:**

I have read many papers in this area.

**Review Assessment: Checking Correctness Of Derivations And Theory:**

I assessed the sensibility of the derivations and theory.

**Review Assessment: Checking Correctness Of Experiments:**

I assessed the sensibility of the experiments.

**Review Assessment: Thoroughness In Paper Reading:**

I read the paper at least twice and used my best judgement in assessing the paper.

---

> ### Author Response · Authors · 2019-11-12
> **Our response to Official Blind Review #2**
>
> Thanks for your constructive comments. We have revised our paper and please kindly check it.
>
> 1. We conducted experiments to compare the computational cost for one Gaussian kernel and the attention similarity map. Given a matrix X containing n examples and each has d dimensional features. We set $n=100$ and record the computational time for different values for $d$, i.e. $d= 100, 200, 300, 400, 500$. The results are reported as follow. For each $d$ value, we run 1000 times and report the average.
> dimension d   ---- 100      -----     200     -----300   -----400       ------500
> Gaussian----------5.84e-4 --   6.62e-4  --6.74e-4   --8.40e-4    --9.57e-4
> Attention---------4.89e-5  --   5.15e-5  --5.59e-5   --7.98e-5    --8.68e-5
> Overall, employing attention could improve the computational cost significantly.
>
> 2. The unary energy is directly computed using GCNs from node features, which means the connectivity of G is explicitly incorporated. In addition, the pairwise is obtained via attention from node features, where we also incorporate the $A^{\ell}$ to consider ${\ell}$-hop connectivity. From our experiments, the dense CRF version always leads to very promising performance. Hence, we believe, generally, increasing ${\ell}$ can lead to better cluster assignments.
>
> 3. We conduct experiments to study how different $k$ values affect the performance. We follow the DGCNN paper [1] to select the number of clusters $k$. Specifically, we use a pooling rate $(0<r<1)$ to control $k$. Then $k$ is set to an integer so that $r*100\%$ of graphs have nodes less than this integer in the current dataset. As suggested in DGCNN, generally, $r=0.9$ is a proper choice for bioinformatics datasets and $r=0.6$ is good for social network datasets. In addition, we conduct experiments to show the performance w.r.t. different $k$ values. We set $r= 0.1, 0.3, 0.5, 0.7, 0.9$ to evaluate the performance on a large-scale social network dataset D&D. The results are reported in Appendix A.4. We observe that the performance drops when $r, k$ is too large or small. In addition, we expect the model to have competitive performance when $r$ is set to a proper range, for example, $r\in[0.3,0.7]$ for dataset D&D.
>
> 4. For the case of different graph features and different graph substructures, it depends on how the GNNs learn and make predictions. It is possible that nodes with different features are clustered in a different way. For example, for the concept network motifs, the nodes in a network motif can be different, but these nodes are combined together to determine the type of the whole graph. In MUTAG dataset, network motifs can be carbon rings, NH2, and NO2, and are highly related to the mutagenic property adn can determine graph label. If the GNNs are trained to make predictions based on network motifs, different nodes in a network motif can be clustered into one cluster. This is our intuitive thought and we believe GNN interpretation techniques are needed if we wish to fully understand the problem.
>
> 5. For the case of repeating instances or repeating instances with different labels, it is related to the data quality. It is a general problem/challenge for all deep models, including our method. If repeating instances have the same label, it can be understood as the unbalanced data problem.
> We can use data preprocessing, data augmentation, weighted loss, or dropout to train better models. If the repeating instances are with different labels and data preprocessing cannot solve it, we believe all GNN models will be affected.
>
> 6. Thanks for the suggestion. We conduct experiments to compare single pooling layer vs multiple pooling layers. Let’s define that 1 block contains 2 GCNs and 1 pooling. Then we compare three different models: 1 block + classifier, 2 blocks + classifier, and 3 blocks + classifier. We evaluate these three models on a large-scale dataset D&D and a small dataset Proteins. The results are reported in Appendix A.3. For the dataset Proteins, we observe that the network with one block can obtain better performance than deeper networks. We believe the main reason is dataset Proteins is a small-scale dataset with an average number of nodes equal to 39.06. A relatively simpler network is powerful enough to learn its data distribution while stacking multiple GCN layers and pooling layers leads to a serious overfitting problem. For the dataset D\&D, the network with 2 blocks performs better than the one with 1 block. Since D\&D is relatively large scale, stacking 2 blocks increases the power of the network and hence increases the performance. However, going very deep, e.g., stacking 3 blocks, will cause the overfitting problem.
>
> 7. Thanks for your suggestion. We rewrote the Section 2.2 to avoid repetition.
>
> [1]. Zhang et al., An End-to-End Deep Learning Architecture for Graph Classification, AAAI 2018

---

### Official Review · AnonReviewer3 · 2019-10-26
**Official Blind Review #3**

**Rating:** 6

**Review:**

The paper proposes a new graph pooling method by learning the node assignments from a CRF based structure prediction formulation. Different from existing pooling methods, the proposed model explicitly captures high-order structural relationships between nodes via a CRF, and the pairwise energy in the CRF is defined by a $l$-hop connection instead of the original neighborhoods. The paper is written clearly, and the experimental results are good. The sensitivity analysis is also complete.

Integrating CRFs with GNNs for node classification is not new ([1][2][3]), but this paper extended energy definition to capture high-order connections and use it for a new subsequent task: graph pooling. It is interesting and novel, but maybe not very significant. (BTW, it is better to add the related works which combined CRF and GNNs, considering the close connection to this paper.)

The sensitivity analysis in the paper is very useful, but it lacks one important ablation study. To justify the usefulness of CRFs instead of other clustering methods for node assignment, I think there should be a baseline which removes the pairwise energy and just use the unary energy for node clustering. Moreover, the paper may also need to compare with a more recent state-of-the-art pooling model SAGPool [4].


[1]Gao, H., Pei, J., & Huang, H. (2019). Conditional Random Field Enhanced Graph Convolutional Neural Networks. In KDD2019.
[2]Ma, T., Xiao, C., Shang, J. and Sun, J., (2018). CGNF: Conditional Graph Neural Fields. https://openreview.net/forum?id=ryxMX2R9YQ
[3]Qu, M., Bengio, Y., & Tang, J. (2019). GMNN: Graph Markov Neural Networks. In ICML2019.
[4]Lee, J., Lee, I., & Kang, J. (2019). Self-Attention Graph Pooling. In ICML 2019.


**Experience Assessment:**

I have published in this field for several years.

**Review Assessment: Checking Correctness Of Derivations And Theory:**

I assessed the sensibility of the derivations and theory.

**Review Assessment: Checking Correctness Of Experiments:**

I assessed the sensibility of the experiments.

**Review Assessment: Thoroughness In Paper Reading:**

I read the paper thoroughly.

---

> ### Author Response · Authors · 2019-11-11
> **Our response to Official Blind Review #3**
>
>
> Thanks for your constructive comments. We have revised our paper and please kindly check it.
>
> 1.Existing methods [1][2][3] incorporate CRFs with GNNs to perform node classification tasks. We include these methods in the related work part, please see the revised manuscript (section 2.3). However, graph classification tasks are also important and require to learn graph level representations. Graph pooling is an effective and efficient way for it, and our proposed method specifically focuses on graph pooling.
>
> 2.We conduct experiments to compare our method with the SAGPool. The results are reported in the Table 2. Experimental results show that our proposed pooling can outperform SAGPool significantly on 5 of 6 datasets. For the dataset Enzymes, the SAGPool performs slightly better than our proposed method (64.17% vs 63.83%). Overall, comparing with SAGPool, our proposed method yields an average of 1.33% performance improvement.
>
> 3.If the pairwise energy is removed, the relations between different node assignments are not explicitly considered. Then the method cannot be considered as a CRF problem. We can remove the pairwise energy and consider it as a baseline but thus it will be similar to Diffpool. All node assignments are obtained by GCNs and their relations are ignored.  Our experimental results are reported in Appendix A.2. By removing the pairwise energy, the performance drops significantly which indicates the importance of pairwise energy.

---

### Author Response · Authors · 2019-11-13
**Our response to all reviewers**

Dear reviewers,
Thanks for the constructive comments.  We have revised our manuscript and please kindly check it. Together with our detailed responses, we hope we have addressed all concerns raised by the reviewers.

---

### Public Comment · ~Siheng_Chen1 · 2020-01-09
**Typos?**

1. In Algorithm 1, Step 5, should 0 ≤ i, j ≤ n? Instead of 0 ≤ i, j ≤ k.
2. In Algorithm 1, Step 8, should it involve addition instead of subtraction in local updating?

---

> ### Author Response · Authors · 2020-01-09
> **Response**
>
> Thanks for pointing out.
> 1. It should be $n$ instead of $k$.
> 2. It is subtraction, not addition. Please refer to the existing work for proof [1][2].
>
> [1]. Zheng et al., Conditional Random Fields as Recurrent Neural Networks, ICCV 2015
> [2]. Krahenbuhl et al., Efficient inference in fully connected crfs with gaussian edge potentials, NIPS 2011

---

### Decision · Program_Chairs · 2019-12-19

**Decision:**

Accept (Poster)

**Comment:**

The paper proposed an operation called StructPool for graph-pooling by treating it as node clustering problem (assigning a label from 1..k to each node) and then use a pairwise CRF structure to jointly infer these labels. The reviewers all think that this is a well-written paper, and the experimental results are adequate to back up the claim that StructPool offers advantage over other graph-pooling operations. Even though the idea of the presented method is simple and it does add more (albeit by a constant factor) to the computational burden of graph neural network, I think this would make a valuable addition to the literature.